# Reliability of the Preliminary OMERACT Juvenile Idiopathic Arthritis MRI Score (OMERACT JAMRIS-SIJ)

**DOI:** 10.3390/jcm10194564

**Published:** 2021-09-30

**Authors:** Tarimobo M. Otobo, Nele Herregods, Jacob L. Jaremko, Iwona Sudol-Szopinska, Walter P. Maksymowych, Arthur B. Meyers, Pamela Weiss, Shirley Tse, Joel Paschke, Rahim Moineddin, Nigil Haroon, Nikolay Tzaribachev, Simone Appenzeller, Olympia Papakonstantinou, Eva Kirkhus, Marion A. J. van Rossum, John Carrino, Philip G. Conaghan, Mirkamal Tolend, Jennifer Stimec, Lennart Jans, Robert G. Lambert, Dax Rumsey, Andrea S. Doria

**Affiliations:** 1Department of Diagnostic Radiology, Hospital for SickKids, Toronto, ON M5G 1X8, Canada; mirkamal.tolend@sickkids.ca (M.T.); jennifer.stimec@sickkids.ca (J.S.); 2Division of Pediatric of Radiology, Universitair Ziekenhuis Ghent, 9000 Ghent, Belgium; nele_herregods@telenet.be (N.H.); lennart.jans@ugent.be (L.J.); 3Department of Radiology and Diagnostic Imaging, University of Alberta, Edmonton, AB T6G 2R3, Canada; jjaremko@ualberta.ca (J.L.J.); rlambert@ualberta.ca (R.G.L.); 4Department of Radiology, National Institute of Geriatrics Rheumatology and Rehabilitation, 02-637 Warsaw, Poland; sudolszopinska@gmail.com; 5Department of Rheumatology, University of Alberta, Edmonton, AB T6G 2R3, Canada; walter.maksymowych@ualberta.ca (W.P.M.); dax.rumsey@albertahealthservices.ca (D.R.); 6Department of Radiology, Cincinnati Children’s Hospital Medical Center, Cincinnati, OH 45229, USA; arthurbmeyers@yahoo.com; 7Department of Rheumatology, Children Hospital of Philadelphia, Philadelphia, PA 19104, USA; weisspa@email.chop.edu; 8Department of Rheumatology, Hospital for SickKids, Toronto, ON M5G 1X8, Canada; shirley.tse@sickkids.ca; 9Care Arthritis Ltd., Edmonton, AB T6W 2Z8, Canada; joel.paschke@carearthritis.com; 10Department of Family Medicine, University of Toronto, Toronto, ON M5G 1V7, Canada; rahim.moineddin@utoronto.ca; 11Department of Rheumatology, Toronto Western Hospital, Toronto, ON M5T 2S8, Canada; nigil.haroon@uhn.ca; 12Department of Pediatric Rheumatology, Medical Center Bad Bramstedt, 24576 Bramstedt, Germany; tzaribachev@pri-research.com; 13Department of Orthopedics, Rheumatology and Traumatology, University of Campinas, Campinas 13083-970, Brazil; appenzellersimone@yahoo.com; 14Department of Radiology, National and Kapodistrian University of Athens, 157-72 Athens, Greece; sogofianol@gmail.com; 15Department of Radiology, Oslo University Hospital, 0372 Oslo, Norway; ekirkhus@ous-hf.no; 16Amsterdam Rheumatology and Immunology Center | Reade and Emma Children’s Hospital Amsterdam University Medical Centers, 1105 Amsterdam, The Netherlands; m.a.vanrossum@amsterdamumc.nl; 17Department of Radiology and Imaging, Hospital for Special Surgery, New York, NY 10021, USA; carrinoj@hss.edu; 18Leeds Institute of Rheumatic and Musculoskeletal Medicine, University of Leeds and NIHR Leeds Biomedical Research Center, Leeds LS7 4SA, UK; p.conaghan@leeds.ac.uk

**Keywords:** OMERACT, JIA, measurement instrument, outcome measure, MRI, SIJ, reliability

## Abstract

This study reports the reliability of the juvenile idiopathic arthritis magnetic resonance imaging scoring system (JAMRIS-SIJ). The study comprised of eight raters—two rheumatologists and six radiologists—and 30 coronal T1 and Short-Tau Inversion Recovery (STIR) MRI scans of patients with enthesitis-related juvenile spondylarthritis. The median age of patients was 15 years with a mean disease duration of 5 years and 22 (73.3%) of the sample were boys. The inter-rater agreement of scores for each of the JAMRIS-SIJ items was calculated using a two-way random effect, absolute agreement, and single rater intraclass correlation coefficient (ICC 2.1). The ICC was interpreted together with kurtosis, since the ICC is also affected by the distribution of scores in the sample. The eight-rater, single measure inter-rater ICC (and kurtosis) values for JAMRIS-SIJ inflammation and damage components were the following: bone marrow edema (BME), 0.76 (1.2); joint space inflammation, 0.60 (1.8); capsulitis, 0.58 (9.2); enthesitis, 0.20 (0.1); ankylosis, 0.89 (35); sclerosis, 0.53 (4.6); erosion, 0.50 (6.5); fat lesion, 0.40 (21); backfill, 0.38 (38). The inter-rater reliability for BME and ankylosis scores was good and met the a priori set ICC threshold, whereas for the other items it was variable and below the selected threshold. Future directives should focus on refinement of the scores, definitions, and methods of interpretation prior to validation of the JAMRIS-SIJ through the assessment of its measurement properties.

## 1. Introduction

Juvenile idiopathic arthritis (JIA) is a chronic inflammatory disease of childhood that affects peripheral and axial joints with onset in a child less than 16 years of age. It is characterized by persistent arthritis for at least 6 weeks and the exclusion of other known conditions. Uncontrolled disease activity has the potential to cause joint damage and growth abnormalities [1,2,3]. Children within the JIA categories including those with enthesitis-related arthritis, arthritis and psoriatic arthritis, and undifferentiated arthritis (often referred to as juvenile spondyloarthritis in case of axial involvement) or juvenile spondyloarthritis (ERA/JSpA) have frequent involvement of their entheses and joints, including the sacroiliac joint (SIJ) [3]. Several JIA disease activity measures exist [4,5,6]; however, their reliability is variable [7], and they often do not represent or include arthritis in axial joints. Clinical examination of the SIJ is a commonly used measure of SIJ disease activity in JIA and is often defined as pain with palpation of the SIJ. However, the validity of clinical assessment is limited by the anatomy and deep location of the SIJ and is not useful in truly differentiating true arthritis from a normal joint in many cases [8,9].

Magnetic resonance imaging (MRI) is considered a valuable non-invasive tool for assessing SIJ inflammation and damage, and for monitoring treatment effectiveness and disease activity in JIA [9,10,11]. Radiography is a frequently used diagnostic imaging technique for SIJ, but its utility is limited because it cannot directly detect early features of disease activity such as bone marrow inflammation [12]. MRI is the most sensitive modality available to assess inflammation in the SIJ [12,13,14]. Reliable and valid MRI imaging instruments have been developed to assess SIJ inflammation and damage in adults [15,16], with special regards to the application of adult joint imaging instruments for the wrist and SIJ in children [1,17,18,19,20]. The need for an imaging outcome measure that was cognizant of the nuances of pediatric bone marrow MRI signal and dynamic bone growth underpinned the development of the preliminary OMERACT juvenile idiopathic arthritis MRI-SIIJ score (JAMRIS-SIJ). This score is a standardized, objective, semi-quantitative, MRI-based outcome measurement instrument developed by a multi-disciplinary international group of experts for the evaluation of SIJ inflammation and structural changes in children with JIA [21]. We aim to assess the inter-rater reliability of this preliminary score in a cohort of children with ERA/JSpA.

## 2. Patients and Methods

Magnetic resonance imaging of boys and girls ≤18 years with confirmed imaging and clinical diagnosis of ERA/JSpA who had a SIJ MRI study performed at The Hospital for Sick Children (Toronto, ON, Canada), the Children’s Hospital of Philadelphia (Philadelphia, PA, USA), or Ghent University Hospital (Ghent, Belgium) comprised the study sample. The MRI examinations were acquired between January 2017 and December 2018. All MR images were anonymized, and patient information was extracted from electronic clinical charts before scoring. All available cases contained at a minimum semicoronal T1-weighted (T1W), T2-weighted (T2W) fat-suppressed, or Short Tau Inversion Recovery (STIR) sequences. Details of sequence protocol are reported in Appendix A. Cases with history of comorbidities such as primary or metastatic bone cancer, SIJ fractures, and not containing the minimum imaging protocol for this study were excluded. The study MRI cases were reviewed for the presence of SIJ pathologies by two pediatric radiologists (JLJ and NH) who were blinded to the clinical history and other imaging findings. The final 30 cases included in the study were randomly selected following a sample size estimation using the Donner and Eliasziw sample size estimation formula for reliability study [22]. They were scored by using the JAMRIS-SIJ scoring methods (Table 1, Figure 1A–C) by 8 raters—2 rheumatologists (one adult and one pediatric rheumatologist) and 6 radiologists (5 pediatric and 1 adult radiologist)—who had between 5 and 33 years of experience assessing pediatric MRIs, (interquartile range (IQR): 10–20, median 14 years).

The JAMRIS-SIJ is a dichotomous item-based instrument that assesses the presence or absence of SIJ inflammation and structural lesions on successive semicoronal T1W, T2W fat-suppressed, and/or STIR sequences through the cartilaginous portion of the SIJ (Table 1). The scores for each item of the JAMRIS-SIJ scale were based on the compilation of information from consecutive semicoronal slices that included the cartilaginous portion of the joint. The definition and scoring methods for the features of the JAMRIS-SIJ scale are described in Table 1 and illustrated in Figure 1A–C. Visualizing a minimum of 1 cm vertical height of the cartilaginous portion of the SIJ is required to meet scoring eligibility. To standardize bone marrow signal for scoring of bone marrow edema (BME), the iliac crest, triradiate cartilage, sacral interforaminal area, and ischiopubic synchondrosis were selected as internal reference comparators for normal pediatric bone marrow signal intensity, and the presacral vein was used as reference for depth.

The inter-rater reliability was calculated by using the two-way random effects model, absolute agreement, and single rater intraclass class correlation coefficient (ICC 2.1) [23]. The reliability estimates assume that both raters and subjects (cases) were randomly selected and represent the population of raters and subjects (cases) from which they were drawn.

Therefore, results can be generalized to other raters and subjects with comparable attributes. The ICCs are reported as absolute agreement for all raters, rheumatologists and radiologists. An a priori determined than an ICC > 0.7 represented an inter-rater reliability estimate, for α = 0.05 and β = 0.20. ICC values were regarded as < 0.5 (poor), 0.5 to 0.75 (moderate), 0.75 to 0.9 (good), and > 0.9 (excellent) reliability [23]. ICC estimates were interpreted together with kurtosis, since the ICC is also affected by the distribution of scores in the sample. Kurtosis is a measure of probability of the tails of a distribution compared to a normal reference kurtosis of 3 (mesokurtic) representing a normal distribution. Kurtosis > 3 (leptokurtic) indicates more distribution of the data in the tail, whereas kurtosis < 3 (platykurtic) refers to a data distribution with a wide bell and less data distribution in the tail.

All raters received initial online training session from an expert developer (NH), comprising examples of SIJ pathologies and how the JAMRIS-SIJ scoring system would be used for the cases, as well as common pitfalls. After the training session, a cognitive debriefing was conducted through email correspondence to clarify any questions or concerns about the scoring method. Lesions were recorded directly on a custom-designed web-based interface (CARE Arthritis) [24] depicting schematics of the SIJ according to SIJ quadrants or halves. Raters were blinded to all patient information except age.

## 3. Results

Out of the 30 patients who had MRI examinations selected for the study 22 (73.3%) were males. The median age of the patients was 14 years (12.3–15.7, range 6–18 years). The degree of disease activity at the time of imaging ranged from minimally active, moderately active, and severe, as 16 (53.3%) cases showed negligible disease burden as reported by the median count of active joints and tender enthesitis of 1, and the remainder of cases showed moderate to severe self-reported pain and a physician global assessment of 5 and 6, respectively. The incidence of JAMRIS-SIJ pathologies among the study MRI examinations were 17 (56%) erosions, 16 (53%) BME; 11 (36.6%) sclerosis, 10 (33.3%) joint space inflammation (JSI), 7 (23.3%) fat lesions; 5 (16.6%) capsulitis, 4 (13.3%) enthesitis, 2 (6.6%) backfill and 1 (3.3%) ankylosis. Seven (23.3%) of the MRI examinations were normal for the patient’s age, exhibiting varying normal growth and age-related variants of the bone marrow signal in the SI region. Descriptive statistics and ICCs for the JAMRIS-SIJ items are reported in Table 2, Table 3 and Table 4. The mean slice count was 8 ± 2 (IQR 7–10), with a minimum of 1 and maximum of 14. Further details on slice counts and mean JAMRIS-SIJ item score among radiologist and rheumatologist are reported in Appendix B and Table 3 and Table 4.

The inter-rater reliability ICCs (and kurtosis) of JAMRIS-SIJ for the inflammation and damage domains were 0.77 (1.0) and 0.60 (6.1), respectively. Among radiologists, the inter-rater reliability estimates were 0.76 (0.48) for the inflammation domain and 0.60 (5.1) for the damage domain, and for the rheumatologists, it was 0.73 (0.84) for the inflammation domain and 0.85 (1.58) for the damage domain.

## 4. Discussion

The reliability of a measurement instrument is a prerequisite for the assessment of its longitudinal and discriminative validity. The final purpose of developing the OMERACT JAMRIS-SIJ was to detect longitudinal change of the SIJ as a post intervention outcome in order to assess treatment efficacy in clinical trials and to quantify axial disease severity in JIA at a given timepoint. Thus, detection of true change is contingent on the optimal reliability of the instrument. This study is the first in the series of validation steps towards evaluating the measurement properties of the JAMRIS-SIJ. Domain-wise inter-rater reliability of the JAMRIS-SIJ achieved the pre-specified threshold for reliability in the study (ICC ≥ 0.7) for inflammation (Table 2, Figure 1D and Figure 2A–C), but not for damage (Table 2, Figure 1D, Figure 2D and Figure 3A–D). Within the inflammation domain, the inter-rater reliability for BME achieved the pre-specified threshold for reliability: however, JSI, capsulitis, and enthesitis did not (Table 2). Within the damage domain, except for ankylosis that reached an ICC ≥ 0.7, the ICCs for sclerosis, erosions, fat lesion and backfill were all < 0.7. The low reliability of the JAMRIS-SIJ structural domain scores (SDS) is consistent with that of other studies [16,19]. Defining an SDS component requires that raters interpret findings based on the assessment of multiple MRI sequences: nevertheless core and optimal protocols for data interpretation have not been agreed upon. This is especially important in children as many patients are small, and the ability to detect subtle findings, such as erosion, may be influenced more by the optimization of the MRI imaging than any disagreement surrounding the definition. At the inception of the JAMRIS-SIJ score, the definitions of structural domain components (SDC) were ambitious. Although consensus was achieved during the development of the SDC definitions, these definitions had minority dissent among the experts. This may have contributed to the low ICC among raters. Of particular importance is the reliability of scoring erosions, since MRI evidence of erosions, BME, and inflammation constitute criteria for the use of biologics in children with JIA [25].

There was a slight difference in ICCs obtained for radiologists and rheumatologists (Table 3 and Table 4), and depending on the JAMRIS-SIJ component, the difference in ICC was either high or low between the specialties (Table 3 and Table 4). Since the scoring method was equally available to all raters, the differences in ICCs may represent different levels of experience of the raters in interpreting musculoskeletal findings in pediatric SIJ MRI [26].

BME is an abnormality that appears in most MRI scoring systems [14]. Its definition had excellent consensus during the development of JAMRIS-SIJ. In this study, inter-rater reliability of BME was consistent, and the good reliability estimates of BME scores may partly be due to the fact that a detailed definition was previously available for scoring of adult and similar description of BME lesions in pediatric SIJs [1,15]. In addition, some of the raters in this study were already familiar with the definition and scoring process of BME. Furthermore, the spectrum of BME abnormalities available in this study may have been limited, which facilitated the decision making of raters during the exercise. Note should be made, however, that the normal spectrum of SIJ signals varies according to the degree of skeletal maturation of children and adolescents [27,28]. As a result, atlases should be developed to address physiologic variation of bone marrow signals across age groups of children and adolescents in future exercises that involve bone marrow scoring in growing joints.

The ICC statistic reflects the proportion of the total variance that is due to variability between subjects in the frequency and extent of the lesion. Consequently, ICC levels will tend to be lower for structural lesions than for BME and applying the same ICC cut-off for what constitutes an acceptable level of reliability may not be appropriate [29]. Notwithstanding the extensive subject content experience and training of most of our study raters, the low ICC performance of some JAMRIS-SIJ items could also be due to the distribution of pathology among cases. There are pragmatic issues with subject selection towards providing a comparable distribution of lesion components represented in each JAMRIS-SIJ domain in opportunity samples such is the case in this study. Noteworthy is the infrequent presence of SIJ ankylosis (an extreme manifestation of JIA) in our study sample. These inherent challenges in the JAMRIS-SIJ measurement component distribution may have adversely affected both subject and component heterogeneity, which, in turn, would have impacted the study inter-rater reliability estimates.

To investigate the non-uniform distribution of our study subjects and components of the JAMRIS-SIJ, the kurtosis of the JAMRIS-SIJ scores was calculated and reported in addition to ICCs. Most JAMRIS-SIJ components were leptokurtic (kurtosis > 3), indicating a substantial deviation of the scores from a uniform distribution. The positive kurtosis alters the subject variance, thereby reducing the reliability estimate. ICC is sensitive to different subject distributions, with optimal ICCs achieved in uniform distributions when rater variability is constant. The values of ICC in our study had a tendency towards under uniformity of distribution as the magnitude of non-uniformity decreased. The overt reliance of ICCs on subject distribution alters the interpretation of ICC as an estimate of agreement in determining the quality of the measurement instrument when the uniformity of the sample subjects and instruments components cannot be guaranteed [29]. These characteristics of the reliability estimates may have contributed to the low reliability in some JAMRIS-SIJ components.

By summing up the component scores of the JAMRIS-SIJ to report the inflammation domain score and structural domain score, equal weight is assumed for each component. However, this may not exactly represent the relative importance of the components in measuring JIA disease activity in the JAMRIS-SIJ inflammation and damage (structural) domains. Furthermore, there may have been overrepresentation of individual components within the domain scores. For instance, within the IDC, the BME item encompasses a significant part of the inflammation domain score (IDS) and consists of three parts: the presence of BME, BME intensity, and BME depth. While BME is a relevant component of the JAMRIS-SIJ, as it signals the beginning of osteochondral inflammation, the appropriate weighting of BME and other components of JAMRIS-SIJ were not considered in this study.

Our study has limitations. Chief among them is the non-uniform case distribution, as noted through the wide variability of score ranges among items of the scale (Table 2, Table 3 and Table 4). Since it was a retrospective study design and the cases were non-randomly selected, we were limited by the availability of cases, making it difficult to achieve a uniform distribution of JAMRI-SIJ pathological lesion. Consequently, uncommon abnormalities such as ankylosis were underrepresented in the study sample, and common abnormalities such as erosions and BME were overrepresented. The significant deviation from uniformity of the case distribution across JAMRIS-SIJ components may have influenced the reliability estimates. Application of sampling methods that reduce the effect of subject distribution on ICC should be adopted in future studies. Further, since this was the first data-driven assessment for inter-rater reliability of the JAMRIS-SIJ item definitions, it was not uncommon for raters to experience challenges to apply the definitions into the scoring methods.

Future directions of research in our study include the utilization of a calibration module for raters and the development of an annotated reference atlas of the JAMRIS-SIJ item abnormalities as a companion measurement aid prior to the next scoring exercise. Moreover, refining the definitions of the JAMRIS-SIJ items based on the challenges encountered in this reliability scoring exercise is critical. Lessons learned from this study will inform steps towards such refinement of the JAMRIS-SIJ items.

## 5. Conclusions

The development of a measurement instrument is an iterative process that follows several steps, comprising of construct definition, selection, definition of measurement items, optimization of scoring methods, conduct of pilot studies, and field testing. This reliability study was a preliminary field testing of the JAMRIS-SIJ, as part of a series of validation processes towards establishing its measurement properties. In this study, we reported the results of the initial inter-rater reliability exercise of the JAMRIS-SIJ in children and adolescents with ERA/JSpA. The JAMRIS-SIJ was originally developed to detect change after treatment intervention (multi-timepoint) and disease severity at a single timepoint in JIA, with special consideration for the unique MRI characteristics of the anatomy of growing SIJs. The JAMRIS-SIJ demonstrated good reliability for the inflammation domain across radiologist and rheumatologist raters. Future steps should aim at the following: further defining the parameters of scoring such as the number of slices scored per reader, improving item-wise scores, item weighting, and item definition refinement; developing a measurement atlas that aligns with the proposed scale; and developing an objective rater calibration and training to improve the JAMRIS-SIJ reliability before proceeding to test its reliability and responsiveness.

## Figures and Tables

**Figure 1 jcm-10-04564-f001:**
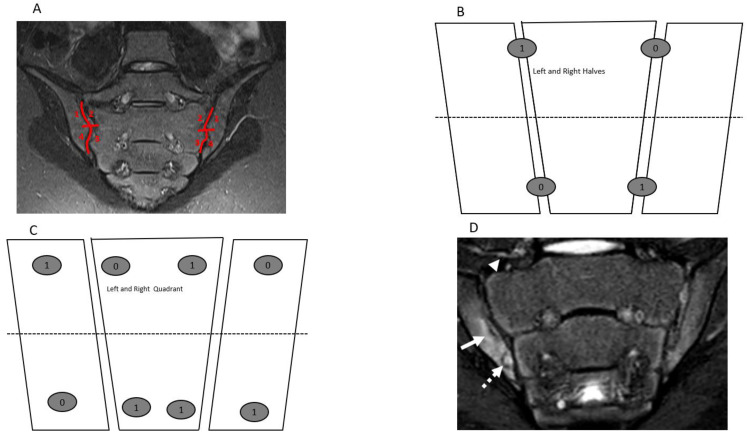
Semicoronal T2 Short Tau Inversion Recovery (STIR) MR images (**A**,**D**) and sketches (**B**,**C**) through the sacroiliac joint (SIJ) illustrate the measurement and method and component of the JAMRIS-SIJ. (**A**) SIJ in a 14-year-old boy shows a normal SIJ divided into quadrants by a vertical line through the joint space and a horizontal line that intersects at the mid portion of the joint (red lines) into superior iliac, sacral, inferior sacral and iliac quadrants in clockwise direction on the right and direction on the left side. (**B**,**C**) Corresponding schema of the SIJ, with (**B**) showing the division of the SIJ into halves and (**C**) showing the division into quadrants with a corresponding example of the scoring JAMRIS-SIJ components. (**D**) SIJ in a 12-year-old boy with enthesitis-related arthritis/juvenile spondylarthritis (ERA/JSPpA) demonstrates bone marrow edema (BME) involving the inferior aspect of the right ilium (**A**—solid arrow). The intensity of the BME signal (arrow) is equal to the signal of the presacral veins. There is a bony defect in the inferior aspect of the right iliac bone seen on both T2-weighted (**A**) and T1-weighted (Figure 2D) images (dashed arrow) with associated BME, consistent with an erosion.

**Figure 2 jcm-10-04564-f002:**
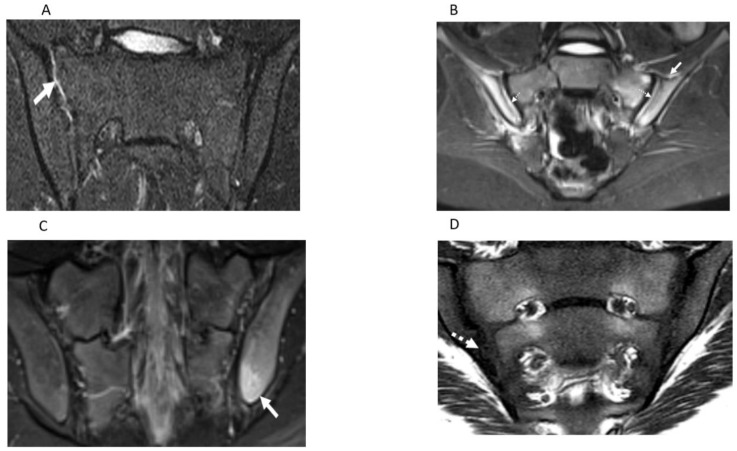
Semicoronal T2 Short Tau Inversion Recovery (STIR) MR images (**A**–**C**) and T1-weighted MR image (**D**) through the sacroiliac joint (SIJ) illustrate measurement components of the JAMRIS-SIJ. (**A**) SIJ in a 17-year-old boy with ERA/JSpA shows joint space inflammation (JSI) as an increased signal intensity within the superior portion of the right SIJ (arrow) compared to the normal signal in the left SIJ. (**B**) SIJ in a 15-year-old boy with ERA/JSpA demonstrates capsulitis as a high signal intensity (arrow) at the superior aspect of the SIJ and JSI (dashed arrow), which is the most pronounced on the right side. (**C**) SIJ in a 17-year-old boy shows edema in the left iliac bone (arrow) inferior and posterior, consistent with enthesitis. (**D**) SIJ in a 12-year-old boy with enthesitis-related arthritis/juvenile spondylarthritis (ERA/JSPpA) demonstrates erosion involving the inferior aspect of the right ilium (dashed solid arrow). The bony defect at the osteochondral interface in the inferior aspect of the right iliac bone corresponds to the location of bone marrow edema (dashed arrow) seen on theT2-weighted sequence of the image (Figure 1D), consistent with an erosion.

**Figure 3 jcm-10-04564-f003:**
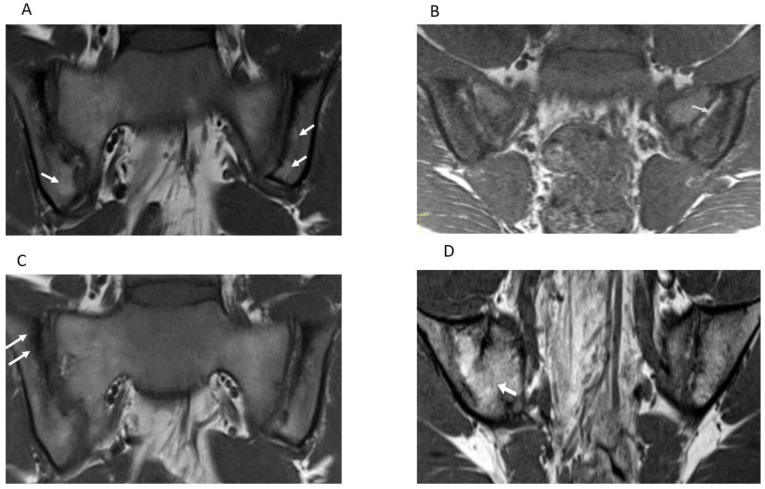
Semicoronal T1-weighted MR images (**A**–**D**) through the sacroiliac joint (SIJ) illustrate measurement components the JAMRIS-SIJ. (**A**) SIJ in a 16-year-old girl with ERA/JSp. (**A**) shows fat lesions as a homogenous increased T1 signal intensity within the inferior iliac subchondral bone (arrows), which is the more prominent on the left. (**B**) SIJ in a 15-year-old boy with ERA/JSpA, shows an area of backfill in the mid to superior subchondral cortex of the left iliac bone. There is increased signal (arrow) that is clearly demarcated from the adjacent normal marrow by irregular dark signal. (**C**) SIJ in a 16-year-old boy with ERA/JSpA shows sclerosis as a low subarticular signal on the superior iliac subchondral bone (arrows). (**D**) SIJ in a 13-year-old boy shows the bone marrow signal extending across the right SIJ (arrow), consistent with ankylosis.

**Table 1 jcm-10-04564-t001:** Features and definitions of inflammation and structural components of an MRI scoring system for sacroiliac joints.

Features	Definitions	Scores
Inflammation MRI Components		Score range/slide
BME	An ill-defined area of high bone marrow signal intensity † within the subchondral bone in the ilium or sacrum on fluid-sensitive images	Score 4 quadrant/SIJ 0/1, range 0–8
BME Intensity	Hyperintensity of the marrow edema using the presacral veins as reference	Score each SIJ 0/1, 0–2
BME Depth	Continues to increase the signal of depth ≥ 5 mm/ ≥ 1 cm from the articular surface	Score each SIJ 0/1, 0–2
Capsulitis	High signal on fluid-sensitive and/or post-contrast enhancement involving the SIJ capsule	Score halves / SIJ 0/1, 0–4
JSI	Increased signal on fluid-sensitive or contrast-enhanced T1-weighted images within the joint space of the cartilaginous portion of the SIJ	Score halves/ SIJ 0/1, 0–4
Enthesitis	High signal in bone marrow and/or soft tissue on a fluid-sensitive sequences or a contrast-enhanced T1-weighted sequence at sites where ligaments and tendons attach to a bone	Score each case 0/1, 0–1
Structural MRI Components		
Sclerosis	A substantially wider than normal area of low subarticular bone signal on T1-weighted and fluid-sensitive images (of ≥5 mm in adolescents)	Score 4 quadrants/SIJ 0/1, 0–8
Erosion	Bony defect (or irregularity with associated bone marrow edema, sclerosis, or fatty lesion) at the osteochondral interface involving both contour and signal on both T1-weighted and fluid-sensitive images	Score 4 quadrants/SIJ 0/1, 0–8
Fat Lesion	Increased homogenous signal intensity on T1-weighted non-FS image in subchondral bone with a distinct border	Score 4 quadrants /SIJ 0/1, 0–8
Backfill	A bright signal on a T1-weighted sequence in a typical location for an erosion, with signal intensity greater than normal bone marrow, and meeting the following requirements. 1. It is associated with complete loss of the dark appearance of the subchondral cortex at its expected location. 2. It is clearly demarcated from adjacent bone marrow by an irregular band dark signal reflecting sclerosis at the border of the original erosion	Score halves/SIJ 0/1, 0–4
Ankylosis	Presence of signal equivalent to regional bone marrow continuously bridging a portion of the joint space between the iliac and sacral bones	Score halves/0/1, 0–4

Statement of overarching consideration for all definitions—“[…] in comparison to physiological changes normally seen in MRIs of age and sex matched children, and visible in two planes where available”. † Caveat for bone marrow edema—“[…] compared to the signal intensity of the iliac crest, edges of the vertebrae, and triradiate cartilage where available”. JAMRIS-SIJ: juvenile arthritis magnetic resonance image sacroiliac joint scoring system; MRI: magnetic resonance imaging; BME: bone marrow edema; SIJ: sacroiliac joint; FS: fat suppressed.

**Table 2 jcm-10-04564-t002:** JAMRIS-SIJ inflammation and damage component descriptive statistic and intraclass correlation coefficients (ICCs) for both radiologists and rheumatologists.

	Bone Marrow Edema	Joint Space Inflammation	Capsulitis	Enthesitis	Inflammation Domain	Sclerosis	Erosion	Fat Lesion	Ankylosis	Backfill	Damage Domain
Mean	17.36	5.48	1.42	0.21	24.26	1.89	6.33	2.49	0.31	0.68	11.70
Median	3	0	0	0	5	0	3	0	0	0	5
SD	24.94	8.30	3.74	0.41	34.26	3.74	9.03	6.31	1.75	3.14	15.54
Kurtosis	1.23	1.75	9.18	0.13	1.06	4.65	6.48	21.11	35.27	38.08	6.14
Skewness	1.48	1.52	3.00	1.46	1.44	2.27	2.19	3.98	5.92	5.93	2.15
Range	99	42	22	1	141	19	60	53	14	27	99
Minimum	0	0	0	0	0	0	0	0	0	0	0
Maximum	99	42	22	1	141	19	60	53	14	27	99
Confidence Level (95.0%)	3.17	1.06	0.48	0.06	4.36	0.48	1.15	0.80	0.22	0.40	1.98
ICC	0.76	0.61	0.58	0.20	0.77	0.54	0.51	0.40	0.90	0.38	0.60

IQR—interquartile range; ICC—intraclass correlation coefficient; SDD—standard deviation.

**Table 3 jcm-10-04564-t003:** JAMRIS-SIJ inflammation components’ descriptive statistic and intraclass correlation coefficients (ICCs) for six radiologists and two rheumatologists.

Radiologist	Bone Marrow Edema	Joint Space Inflammation	Capsulitis	Enthesitis	Inflammation Domain	Rheumatologist	Bone Marrow Edema	Joint Space Inflammation	Capsulitis	Enthesitis	Inflammation Domain
Mean	19.87	5.84	1.64	0.21	27.35	Mean	9.85	4.38	0.75	0.08	14.98
Median	4	0	0	0	7	Median	1	0	0	0	1
SD	26.98	8.59	4.09	0.40	36.79	SD	15.28	7.30	2.27	0.28	23.03
Kurtosis	0.52	1.78	7.48	0.04	0.49	Kurtosis	1.16	0.76	9.74	7.83	0.84
Skewness	1.28	1.51	2.77	1.43	1.29	Skewness	1.54	1.478	3.23	3.09	1.48
Range	99	42	22	1	141	Range	56	24	11	1	80
Confidence level (95.0%)	3.97	1.26	0.60	0.06	5.41	Confidence Level (95.0%)	3.95	1.89	0.59	0.07	5.95
IQR	0–35	0–11	0–0	0–0	0–50	IQR	0–14	0–6	0–0	0–0	0–21
ICC	0.82	0.61	0.60	0.16	0.81	ICC	0.76	0.50	0.80	0.09	0.73

IQR—interquartile range; ICC—intraclass correlation coefficient; SDD—standard deviation.

**Table 4 jcm-10-04564-t004:** JAMRIS-SIJ damage components’ descriptive statistic and intraclass correlation coefficients (ICCs) for six radiologists and two rheumatologists.

Radiologist	Sclerosis	Erosion	Fatty Lesion	Ankylosis	Backfill	Damage Domain	Rheumatologist	Sclerosis	Erosion	Fatty Lesion	Ankylosis	Backfill	Damage Domain
Mean	1.93	6.66	3.19	0.28	0.72	12.77	Mean	1.75	5.37	0.38	0.40	0.57	8.47
Median	0	3	0	0	0	6	Median	0	2.5	0	0	0	4.5
SD	3.88	9.58	7.10	1.57	3.23	16.87	SD	3.30	7.09	1.53	2.20	2.87	10.04
Kurtosis	4.72	6.47	15.77	34.49	36.95	5.16	Kurtosis	3.55	0.20	22.47	30.85	45.25	1.58
Skewness	2.30	2.24	3.44	5.88	5.81	2.04	Skewness	2.05	1.22	4.66	5.57	6.49	1.34
Range	19	60	53	11	27	99	Range	14	24	9	14	21	44
Confidence level (95.0%)	0.57	1.41	1.04	0.23	0.48	2.48	Confidence Level (95.0%)	0.85	1.83	0.40	0.57	0.74	2.59
IQR	0–2	0–10	0–4	0–0	0–0	0–20	IQR	0–2	0–9	0–0	0–0	0–0	0–15
ICC	0.61	0.44	0.52	0.90	0.29	0.62	ICC	0.17	0.17	0.02	0.95	0.53	0.86

IQR—interquartile range; ICC—intraclass correlation coefficient; SDD—standard deviation.

## Data Availability

Data presented in this study are available on request from the corresponding author.

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
