# Peer review of "Reliability of the Preliminary OMERACT Juvenile Idiopathic Arthritis MRI Score (OMERACT JAMRIS-SIJ)"

_jcm, 2021, doi:10.3390/jcm10194564_

Round 1

Reviewer 1 Report

Title: Reliability of the Preliminary OMERACT Juvenile Idiopathic Arthritis MRI Score (OMERACT JAMRIS SIJ)

An interesting study focusing on assisting inter-rater reliability of a new MRI scoring system for enthesitis-related JIA. Some parts of the paper require greater clarity.

Abstract:

  1. Method: Please give some more information about who study population was e.g. how many scans from how many patients. Brief information about average age, disease duration.
  2. Method: By juvenile spondyloarthritis, do you mean enthesitis-related JIA? Please clarify in the text which JIA categories were included.

Introduction

  1. Line 53: Arthritis and psoriasis – please clarify in the text that you are referring to psoriatic JIA (this category of JIA does not actually require arthritis and psoriasis in the presence of other factors e.g. family history)

Patients and Methods

  1. Line 83: Please clarify the time period for data collection in the study. Were MRI images kept from any appointment from any child with ERA who was seen between Jan 2017 and Dec 2018, or was this when the study investigators extracted data from previous records e.g. from children seen at a much earlier time point?
  2. Please clarify whether only one image per child was kept or whether all MRI images were included even when multiple per child. Did you consider testing intra-rater reliability where multiple images were taken?
  3. Line 92: It is unclear whether the candidate study cases and the final 30 cases are part of the same sample or additional cases. Were the 30 cases selected at random from the candidate study cases, or were these additionally included from clinical records as the sample size was not met?
  4. Line 122: All raters received online training. Do you envisage that everyone who uses this tool will need the same online training and is this feasible to implement in a larger scale?
  5. Line 129: Why did you not blind the raters to the age of the participants?
  6. Did you consider objectively comparing the reliability of scoring between radiologists and rheumatologists as two groups on the same images?

Results:

  1. It would be interesting to include some information about people who could not be included in the study. E.g. how many patients with ERA were seen at the hospitals in the study period, how many had MRI scans, and of those, how many could be included? This is important as if only a select number of MRI scans were of quality to use this tool, it affects the applicability in a clinical setting/for future research.
  2. Line 132: Degree of disease activity is reported but how this was measured is not described in the methods.
  3. Line 136: Are the pain and physician global assessment scores of 5 and 6 in centimetres as reported on the visual analogue scales? Please clarify in methods and results.
  4. Line 146: Many readers will not be aware how to interpret kurtosis values. The authors have well-explained how to interpret ICC values, please also include a sentence in the methods how kurtosis values should be interpreted.

Table 2:

  1. Is the confidence level reported that surrounding the ICC? If yes, please the upper and lower confidence limits.

Author Response

Response to Reviewer 1

Abstract:

  1. Method: Please give some more information about who the study population was e.g., how many scans from how many patients. Brief information about average age, disease duration.

Response:  A total of 30 coronal T1 and Short Tau Inversion Recovery (STIR) MRI scans from 30 patients, with Enthesitis Related Arthritis – Juvenile Idiopathic Arthritis (ERA/JIA) comprised the study sample. The median age of the patient was 15 years with a mean disease duration of 5 years and 22 (73.3%) of the sample were boys.

  1. Method: By juvenile spondyloarthritis, do you mean enthesitis-related JIA? Please clarify in the text which JIA categories were included.

Response:  The category of JIA of patients in the study was enthesitis-related arthritis/JIA

Introduction

  1. Line 53: Arthritis and psoriasis – please clarify in the text that you are referring to psoriatic JIA (this category of JIA does not actually require arthritis and psoriasis in the presence of other factors e.g., family history)

Response:  We meant psoriatic arthritis

Patients and Methods

  1. Line 83: Please clarify the time period for data collection in the study. Were MRI images kept from any appointment from any child with ERA who was seen between Jan 2017 and Dec 2018, or was this when the study investigators extracted data from previous records e.g., from children seen at a much earlier time point?

Response:  This was the period investigators extracted the data from previous records.

  1. Please clarify whether only one image per child was kept or whether all MRI images were included even when multiple per child. Did you consider testing intra-rater reliability where multiple images were taken?

Response:   Only one set of scans at one-time point was selected for each patient. Slices were read using the study protocol and only slices that met the inclusion criteria were scored by experts. Intra-rater reliability was not estimated.  

  1. Line 92: It is unclear whether the candidate study cases and the final 30 cases are part of the same sample or additional cases. Were the 30 cases selected at random from the candidate study cases, or were these additionally included from clinical records as the sample size was not met?

Response:  The candidate study cases and final cases are the same. The 30 cases were a convenient sample of MRI studies that were selected from the PACS of the case contributing centers.

  1. Line 122: All raters received online training. Do you envisage that everyone who uses this tool will need the same online training and is this feasible to implement in a larger scale?

Response:  Everyone who uses the JAMARIS-SIJ requires calibration, as the reliability the dependent on content expertise. Since the calibration is easily accessible online, it can easily be accessible to potential users of the tool and feasible to implement in large-scale clinical trial studies.

  1. Line 129: Why did you not blind the raters to the age of the participants?

Response:  there is evidence of age-related normal MRI features that could mimic ERA findings in the SIJ. Providing the age to the rater controls the pitfall in the interpretation of the cases, as the final interpretation will be made cognizant of the age of the patient. ( Chauvin, N. A. et al. MRI of the Sacroiliac Joint in Healthy Children. AJR Am. J. Roentgenol., 1-7, doi:10.2214/ajr.18.20708 (2019). And Herregods, N. et al. Normal subchondral high T2 signal on MRI mimicking sacroiliitis in children: frequency, age distribution, and relationship to skeletal maturity. Eur. Radiol., doi:10.1007/s00330-020-07328-0 (2020)).

  1. Did you consider objectively comparing the reliability of scoring between radiologists and rheumatologists as two groups on the same images?

Response:  The reliability of the radiologist and rheumatologist were reported but we were cautious to make the comparison as the reader sample was not equal. More so, if reader calibration is archived, the effect of a reader specialty on the reliability may be negligible or absent. However, since this was a preliminary study, subgroup differences of the readers were not a priority.

Results:

  1. It would be interesting to include some information about people who could not be included in the study. E.g. how many patients with ERA were seen at the hospitals in the study period, how many had MRI scans, and of those, how many could be included? This is important as if only a select number of MRI scans were of a quality to use this tool, it affects the applicability in a clinical setting/for future research.

Response:  The MRI studies were selected from the picture archiving and communication system of the case contributing centers. As this was a retrospective data collection, we do not preview to information about patient utilization rate at the various centers. The study protocol has clearly defined inclusion criteria for MRI scans that will be included in the study.

  1. Line 132: Degree of disease activity is reported but how this was measured is not described in the methods.

Response:  Disease activities are an outcome that is measure from a constellation of clinical assessments, these included the active joint count and tender enthesitis, patient-reported or proxy reported pain, and physician global assessment. These parameters were reported from the clinical charts of the patients included in the study.

  1. Line 136: Are the pain and physician global assessment scores of 5 and 6 in centimeters as reported on the visual analog scales? Please clarify in methods and results.

Response: They are numerical arithmetic counts and do not have metrics.

  1. Line 146: Many readers will not be aware of how to interpret kurtosis values. The authors have well-explained how to interpret ICC values, please also include a sentence in the methods how kurtosis values should be interpreted.

Response: Kurtosis is a measure of the probability of the tails of a distribution compared to a normal reference kurtosis of 3 (mesokurtic) representing a normal distribution. Kurtosis > 3 (leptokurtic) indicates more distribution of the data in the tail, whereas kurtosis < 3 (platykurtic) refers to a data distribution with a wide bell and less data distribution in the tail.

Table 2:

  1. Is the confidence level reported that surrounding the ICC? If yes, please the upper and lower confidence limits.

 Response: The confidence intervals were not reported for the ICC

Reviewer 2 Report

The article "Reliability of the Preliminary OMERACT Juvenile Idiopathic Arthritis MRI Score (OMERACT JAMRIS-SIJ)" represent the OMEACT aspects which are in line with all the scientific specifications firstly formulated at the first OMERACT conference held in Maastricht, The Netherlands in 1992 and its amendments at a later period.

The peer-reviewer would like to highlight some aspects, which might be considered by the authors, but also it does not lower the scientifically-based subject of the article. 

These aspects, which are mentioned in No. 1 and No. 2 would be appreciated to discuss shortly in the appropriate body of the proposed article if the authors consider this relevant to the reader. 
On the other hand, it is highly recommended that aspect No. 3 should be unambiguously discussed in the article body.

No. 1:
 The peer-reviewer would like to know the authors' opinion and to discuss briefly the potential/possible future impact on:
- clinical praxis in terms of the therapy of Juvenile Idiopathic Arthritis,
- the potential impact on future randomised clinical trials for a new therapies development, 
- the potential impact on public health (in comparison with the state-of-the-art of the general domains of health status defined by the "D's")

No. 2: 
The peer-reviewer would like to know the authors' opinion and to discuss briefly the feasibility of this study, i.e. can the measure be applied easily, given constraints of time, cost, and interpretability? The feasibility captures an essential element in the selection of measures, one that in the end may be decisive in determining a measure's success in the future context of directives focused on refinement of the scores, definitions, and methods of interpretation prior to validation of the JAMRIS-SIJ through assessment of its measurement properties. In this context please consider the statistical power (the number of paediatric subjects involved) in your study. The peer-reviewer is aware, this is a preliminary cohort study, however, others will build and try to establish better approaches on your preliminary results. Consequently, the maximum accuracy of the results obtained and the statistical interpretation is highly recommended.

No. 3:
- The peer-reviewer would like to ask the authors to discuss some uncertainties in Table 3 and Table 4. The abbreviation "SD" there in both tables cannot be without a doubt considered as Standard Deviation as it has not been explained in the legend below. 
If the peer-reviewer consider the abbreviation "SD" as Standard Deviation, the question to the authors is, why the presentation of the numerical values for Radiologists and Rheumatologists are the same in Table 3 and the same in Table 4 for "SD" line? Otherwise justified, if the abbreviation "SD" should refer to something else, all this should be mentioned and discussed in legend and results.
- There in the body of the whole article, Table 4 is isolated and presented without any further comments and connection to the study described. The authors are asked to clearly point the results from Table 4 to the appropriate context in the text for the results and discussion section.
- If relevant, Table 1 should be discussed and clearly pointed in the discussion section as well.
- Figures 1, 2 and 3 are already mentioned in the discussion section with an appropriate cross-reference (the pictures vs text). However, there in the results section the appropriate description of the results from Figures 1, 2 and 3 is missing. The authors are asked to discuss the omission of this aspect in section results of the proposed article.

Author Response

Response to Comments and Suggestions for Authors

The article "Reliability of the Preliminary OMERACT Juvenile Idiopathic Arthritis MRI Score (OMERACT JAMRIS-SIJ)" represent the OMEACT aspects which are in line with all the scientific specifications firstly formulated at the first OMERACT conference held in Maastricht, The Netherlands in 1992 and its amendments at a later period.

The peer-reviewer would like to highlight some aspects, which might be considered by the authors, but also it does not lower the scientifically based subject of the article. 

These aspects, which are mentioned in No. 1 and No. 2 would be appreciated to discuss shortly in the appropriate body of the proposed article if the authors consider this relevant to the reader. 
On the other hand, it is highly recommended that aspect No. 3 should be unambiguously discussed in the article body.

No. 1:
 The peer-reviewer would like to know the authors' opinion and to discuss briefly the potential/possible future impact on:
- clinical praxis in terms of the therapy of Juvenile Idiopathic Arthritis,
- the potential impact on future randomised clinical trials for a new therapies development, 
- the potential impact on public health (in comparison with the state-of-the-art of the general domains of health status defined by the "D's")

Response: The authors are reserved about making a recommendation at a preliminary stage of the scale development. Potential values for imaging outcome measures such as the JAMRIS-SIJ have been highlighted in the introduction.

No. 2: 
The peer-reviewer would like to know the authors' opinion and to discuss briefly the feasibility of this study, i.e., can the measure be applied easily, given constraints of time, cost, and interpretability? The feasibility captures an essential element in the selection of measures, one that in the end may be decisive in determining a measure's success in the future context of directives focused on refinement of the scores, definitions, and methods of interpretation prior to validation of the JAMRIS-SIJ through assessment of its measurement properties. In this context, please consider the statistical power (the number of paediatric subjects involved) in your study. The peer-reviewer is aware, this is a preliminary cohort study, however, others will build and try to establish better approaches on your preliminary results. Consequently, the maximum accuracy of the results obtained, and the statistical interpretation is highly recommended.

Response: The feasibility of the JAMRIS-SIJ was not evaluated as the current scoring system is undergoing significant revision following the results of this study. The feasibility of a scoring system is valuable but was not an objective in this study.

No. 3:
- The peer-reviewer would like to ask the authors to discuss some uncertainties in Table 3 and Table 4. The abbreviation "SD" there in both tables cannot be without a doubt considered as Standard Deviation as it has not been explained in the legend below. 
If the peer-reviewer consider the abbreviation "SD" as Standard Deviation, the question to the authors is, why the presentation of the numerical values for Radiologists and Rheumatologists are the same in Table 3 and the same in Table 4 for "SD" line? Otherwise justified, if the abbreviation "SD" should refer to something else, all this should be mentioned and discussed in legend and results.
- There in the body of the whole article, Table 4 is isolated and presented without any further comments and connection to the study described. The authors are asked to clearly point the results from Table 4 to the appropriate context in the text for the results and discussion section.
- If relevant, Table 1 should be discussed and clearly pointed in the discussion section as well.
- Figures 1, 2 and 3 are already mentioned in the discussion section with an appropriate cross-reference (the pictures vs text). However, there in the results section the appropriate description of the results from Figures 1, 2 and 3 is missing. The authors are asked to discuss the omission of this aspect in section results of the proposed article.

Response: Track changes have been made to the table and description to comments on all tables in the manuscript. SD stands for standard deviation, and the values are different for radiologists and rheumatologists. There was a data entry mistake that has been correct in Tables 3 and 4 as uploaded in the tracked changed document.